# VMFTransformer: An Angle-Preserving and Auto-Scaling Machine for Multi-horizon Probabilistic Forecasting

## Abstract

Time series forecasting has historically been a key area of academic research and industrial applications. As deep learning develops, the major research methodologies of time series forecasting can be divided into two categories, i.e., iterative and direct methods. In the iterative methods, since a small amount of error is produced at each time step, the recursive structure can potentially lead to large error accumulations over longer forecasting horizons. Although the direct methods can avoid this puzzle involved in the iterative methods, it faces abuse of conditional independence among time points. This impractical assumption can also lead to biased models. To solve these challenges, we propose a direct approach for multi-horizon probabilistic forecasting, which can effectively characterize the dependence across future horizons. Specifically, we consider the multi-horizon target as a random vector. The direction of the vector embodies the temporal dependence, and the length of the vector measures the overall scale across each horizon. Therefore, we respectively apply the von Mises-Fisher (VMF) distribution and the truncated normal distribution to characterize the angle and the magnitude of the target vector in our model. We evaluate the performance of our framework on three benchmarks. Extensive results demonstrate the superiority of our framework over six state-of-the-art methods and show the remarkable versatility and extensibility for different time series forecasting tasks.

## 1 Introduction

Time series forecasting has historically been a key area of academic research and industrial applications, such as climate modeling (Mudelsee, 2019), biological sciences (Stoffer & Ombao, 2012), medicine (Topol, 2019), detail decision making (Böse et al., 2017), and finance (Andersen et al., 2005). The common requirement of time series forecasting is measuring the uncertainty of the output by predicting its probability distribution, which is termed "probabilistic forecasting". In addition, the practical use of probabilistic forecasting generally requires forecasting more than one step, i.e., multi-horizon forecasting. For example, in retail, multi-horizon probabilistic forecasting is used for optimal inventory management, staff scheduling, and topology planning (Simchi-Levi et al., 2008); in finance, it is used to prevent sudden flow abnormalities (Balbás et al., 2005). Modern machine learning methods have been proposed for multi-horizon probabilistic forecasting, which can be divided into iterative and direct methods as follows.

Iterative approaches typically make use of autoregressive deep learning architectures, which produce multi-horizon forecasts by recursively feeding samples of the target into future time steps. Iterative approaches generally make use of the "chain rule" in the training stage, which decomposes $p(\mathbf{y}_{T+1:T+H}|\mathbf{y}_{0:T}, \mathbf{x})$ as $\prod_{t=T+1}^{T+H} p(y_t|\mathbf{y}_{0:t-1}, \mathbf{x})$ (Sutskever et al., 2014), where $\mathbf{y}_{0:T} = (y_0, y_1, \ldots, y_T)^T$ denotes a slice of the time series and $\mathbf{x}$ represents some external features. Due to the chain rule, iterative approaches transform the estimation of $p(\mathbf{y}_{T+1:T+H}|\mathbf{y}_{0:T}, \mathbf{x})$ into a one-step-ahead prediction in the training stage and feed the prediction of $y_{t-1}$ back as ground truth to forecast $y_t$. However, as pointed out in (Bengio et al., 2015; Lamb et al., 2016; Wen et al., 2017), the discrepancy between actual data and estimates during prediction can lead to error accumulation. Since a small amount of error is produced at each time step, the recursive structure of iterative methods can potentially lead to large error accumulations over long forecasting horizons. Therefore, the

iterative approaches are less robust and might lead to a biased model (Chevillon, 2007; Taieb & Atiya, 2016).

For the direct methods (Wen et al., 2017; Lim et al., 2021; Fan et al., 2019), they can alleviate the above-mentioned issues involved in iterative methods, which can directly forecast all targets by using all available inputs. They typically use sequence-to-sequence architecture, a type of network structure mapping sequences directly to sequences. These methods commonly choose the quantile loss function as the training objective. They try to jointly minimize the quantile loss at each future horizon, where a decoder structure propagates encoded historical information and processes external features that can be acquired in advance. However, minimizing a loss function at each horizon, which is equivalent to transforming $p(\mathbf{y}_{T+1:T+H}|\mathbf{y}_{0:T}, \mathbf{x})$ into $\prod_{h=1}^{H} p(y_{T+h}|\mathbf{y}_{0:T}, \mathbf{x})$, is somewhat abuse of conditional independence. The discrepancy between $p(\mathbf{y}_{T+1:T+H}|\mathbf{y}_{0:T}, \mathbf{x})$ and $\prod_{h=1}^{H} p(y_{T+h}|\mathbf{y}_{0:T}, \mathbf{x}, h)$ can also lead to biased models.

Taking all the aforementioned challenges into account, we propose a direct approach for multi-horizon probabilistic forecasting which can effectively capture the characteristics of the dependence across future horizons. We consider the multi-horizon target $\mathbf{y}_{T+1:T+H}$ as a random vector in an $H$-dimension vector space. When there exists a certain dependence mechanism among $y_{T+h}$ ($1 \leq h \leq H$), $\mathbf{y}_{T+1:T+H}$ is likely to be distributed around a specific direction. The direction of $\mathbf{y}_{T+1:T+H}$ can be defined by the cosine of its angles relative to the orthonormal basis $\{e_h\}_{h=1}^{H}$ [1] of the vector space, which is an $H$-dimension unit-vector defined as

$$\left( \frac{\langle \mathbf{y}_{T+1:T+H}, e_1 \rangle}{||\mathbf{y}_{T+1:T+H}||_2}, \ldots, \frac{\langle \mathbf{y}_{T+1:T+H}, e_H \rangle}{||\mathbf{y}_{T+1:T+H}||_2} \right)^T,$$

where $\langle, \rangle$ denotes the inner product and $||\mathbf{y}_{T+1:T+H}||_2 = \sqrt{\sum_{h=1}^{H} y_{T+h}^2}$ denotes the Euclidean norm of $\mathbf{y}_{T+1:T+H}$. Therefore, we apply the von Mises-Fisher (VMF) distribution, which is a probability distribution on the surface of a unit-sphere, to characterize the distribution of the direction of $\mathbf{y}_{T+1:T+H}$. Once the direction of $\mathbf{y}_{T+1:T+H}$ is learned, and suppose $||\mathbf{y}_{T+1:T+H}||_2$ is given, the forecast can be made by multiplying $||\mathbf{y}_{T+1:T+H}||_2$ with its direction. Hence, we normalize $\mathbf{y}_{T+1:T+H}$ by dividing its length and adopt a prior distribution on $||\mathbf{y}_{T+1:T+H}||_2$ to obtain a complete tractable likelihood function. Recall that the direction of $\mathbf{y}_{T+1:T+H}$ is defined via angles, and its length is determined by the scale of each $y_{T+h}$, the similarity measurement of the attention module involved in our model is correspondingly modified, which is capable of evaluating the joint similarity between angles and scales, namely the "Angle&Scale" similarity. The key features of our method are to preserve the temporal dependence, or as we explained before, the angles of $\mathbf{y}_{T+1:T+H}$, and automatically project $||\mathbf{y}_{T+1:T+H}||_2$ to each $y_{T+h}$ to estimate its scale. We summarize these features as "angle-preserving" and "auto-scaling", where auto-scaling is important for handling data with different magnitudes (Salinas et al., 2019). One remaining challenge is optimizing the likelihood function due to the Bessel function, an essential part of VMF distribution and commonly leading to underflow problems(Kumar & Tsvetkov, 2018). Without the sacrifice of accuracy, we estimate and alternatively optimize the upper bound of the Bessel function.

Our contributions are three folds: (1) We propose a probabilistic forecasting model (VMFTransformer) based on Transformer and VMF distribution, which captures the temporal dependence of multi-horizon targets. We also demonstrate our model's performance is state-of-the-art on real-world datasets; (2) We design a novel similarity measurement termed "Angle&Scale" similarity for the attention module; (3) We present a more efficient optimization method for the Bessel function in the VMF distribution without the sacrifice of accuracy.

## 2 RELATED WORK

**Time Series Forecasting.** Recent time series forecasting models based on deep learning (e.g., recurrent and convolutional neural networks) (Salinas et al., 2019; Rangapuram et al., 2018; Wen et al., 2017) provide a data-driven manner to deal with time series forecasting tasks and achieve great accuracy in most application fields. Due to complex dependencies over time of recurrent networks and the limits of convolutional filters, these methods have difficulties in modeling long-term and complex relations in the time series data.

---

[1] $e_h = (0, \ldots, 1, \ldots, 0)$, the $h$-th element is 1 while the rest are 0.

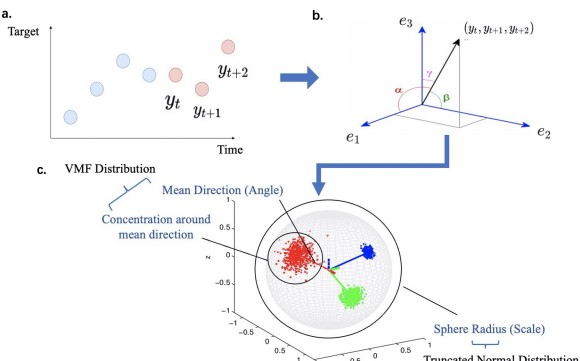

Figure 1: Illustration of how a multi-horizon target can be viewed as a high-dimensional random vector and how the VMF distribution is applied to describe its distribution. As shown in Figure 1.a, the example shows when performing a 3-step forecasting, the target can be packed together into a 3-dimensional vector $(y_t, y_{t+1}, y_{t+2})$ where its direction is a unit-vector, i.e. $(\cos\alpha, \cos\beta, \cos\gamma)$, determined by its angles relative to the orthonormal basis $\{e_1, e_2, e_3\}$ in Figure 1.b. When there exists dependency among $y_t$, $y_{t+1}$, and $y_{t+2}$, the random vector $(y_t, y_{t+1}, y_{t+2})$ is likely to be distributed around a certain direction, described by the VMF distribution for random vectors on unit-sphere in Figure 1.c. The scale or length of $(y_t, y_{t+1}, y_{t+2})$ is equivalent to sphere radius, whose distribution is described by the Truncated Normal Distribution in Figure 1.c. A forecast can be made by multiplying a direction generated from the estimated VMF distribution with a length generated from the estimated Truncated Normal Distribution.

Recently, Transformers (Vaswani et al., 2017) based on self-attention mechanism (Fan et al., 2019) show promising performance in time series forecasting (Li et al., 2020; Wu et al., 2021; Zhou et al., 2021; Kitaev et al., 2020). Considering the dependencies of each time point in a sequence, Transformer-based methods (Li et al., 2020) are proposed by assigning different importance to the different time points. Progresses have been made in reducing the computation complexity of the self-attention and enhancing the capacity of information extraction of the Encoder-Decoder structure(Zhou et al., 2021; Wu et al., 2021; Kitaev et al., 2020). In addition, Matrix factorization methods (Yu et al., 2016) and Bayesian methods that share information via hierarchical priors (Chapados, 2014) are used to learn multiple related time series by leveraging hierarchical structure (Hyndman et al., 2011). Despite the empirically extraordinary performance, these iterative and direct methods are all still facing the mentioned challenges in Introduction.

**von Mises-Fisher (VMF) distribution.** The von Mises Fisher Distribution (VMF) is an important isotropic distribution for directional data that have direction as well as magnitude, such as gene expression data, wind current directions, or measurements taken from compasses (Dhillon & Sra, 2003). The von Mises–Fisher distribution is a probability distribution on directions in $\mathbb{R}^p$. It can be regarded as a distribution on the $(p-1)$-sphere of unit radius, which is on the surface of the D-ball of unit radius. If $p = 2$, the distribution reduces to the von Mises distribution on the circle. Recently, it has been successfully used in numerous machine learning tasks, such as unsupervised learning (Banerjee et al., 2005; Gopal & Yang, 2014), supervised learning (Scott et al., 2021), contrastive learning (Wang & Isola, 2020), natural language processing (Kumar & Tsvetkov, 2018), computer vision (Hasnat et al., 2017; Zhang et al., 2021), and so on. Our work is the first to introduce the von Mises-Fisher distribution to the time series forecasting task.

## 3 METHODOLOGY

The core idea of this work is to consider a multi-horizon target as a vector in a high-dimensional space, where its direction or angle relative to the orthonormal basis characterizes the dependence structure, and its length is determined by the scale of the targets at each horizon. Therefore, performing probabilistic forecasting requires first evaluating the probability distribution of directions. For this purpose, we appeal to the VMF distribution. Recall that the VMF distribution only applies to vectors distributed on a unit sphere; we, therefore, adopt a probability distribution on the length of the vector representing the multi-horizon target. We choose the truncated normal distribution for vector length, which turns out to be a prior distribution. We visualize this idea in Figure 1.

The rest of this section is organized into four subsections. We first derive the objective function for model training and introduce the attention module to suit our proposed objective function. We then propose a training trick to increase the stability of gradient descent while presenting a random sampling method for performing probabilistic forecasts.

### 3.1 OBJECTIVE FUNCTION

This section derives the log-likelihood function for maximum likelihood estimation (MLE) at the training stage.

**VMF Distribution**. The VMF distribution is a probability distribution on the surface of a unit-sphere. For a $d$ dimensional random unit vector $\mathbf{y} = (y_1, \ldots, y_d)^T$ ( $||\mathbf{y}||_2 = 1$ ), the probability density function of VMF distribution is defined as

$$p(\mathbf{y}; \boldsymbol{\mu}, \kappa) = C_d(\kappa) \exp(\kappa \boldsymbol{\mu}^T \times \mathbf{y}), \tag{1}$$

where $\boldsymbol{\mu}$ denotes the mean direction ($||\boldsymbol{\mu}||_2 = 1$), and $\kappa$ denotes the concentration parameter. In other words, $\boldsymbol{\mu}$ locates the most likely direction of $\mathbf{y} = (y_1, \ldots, y_d)^T$, and $\kappa$ controls the divergence of $\mathbf{y} = (y_1, \ldots, y_d)^T$ from $\boldsymbol{\mu}$. The greater the value of $\kappa$, the stronger concentration of $\mathbf{y} = (y_1, \ldots, y_d)^T$ around $\boldsymbol{\mu}$. The normalization constant $C_d(\kappa)$ in Equation equation 1 is defined as

$$C_d(\kappa) = \frac{\kappa^{d/2-1}}{(2\pi^{d/2})I_{d/2-1}(\kappa)},$$

where $I_{d/2-1}(\kappa)$ [2] is the modified Bessel function of the first kind.

**Conditional Density Function**. Let $\sigma$ denote the length of the multi-horizon target, i.e. $||\mathbf{y}_{T+1:T+H}||_2 = \sigma$, based on Equation equation 1, we define the conditional probability density function of $\mathbf{y}_{T+1:T+H}$ as

$$p(\mathbf{y}_{T+1:T+H}|\sigma; \boldsymbol{\mu}, \kappa) = C_H(\kappa) \exp\left(\kappa \boldsymbol{\mu}^T \times \frac{\mathbf{y}_{T+1:T+H}}{\sigma}\right). \tag{2}$$

**Prior on $\sigma$**. We introduce a prior distribution on the scale parameter $\sigma$ to make Equation equation 2 tractable. For explicitness, we choose the truncated normal distribution (Burkardt, 2014). The truncated normal distribution is an extension of the normal distribution, which compress the range of a random variable from $(-\infty, +\infty)$ into an open interval $(a, b)$ $(-\infty \le a < b \le +\infty)$. A truncated normal distribution is determined by four parameters $m, \gamma, a, b$. The parameters $m, \gamma$ denote the location and shape parameters, respectively, while $a, b$ denote the lower and upper bound of the random variable. In our case, $a = 0$ and $b = +\infty$. Technically,

$$p(\sigma^2; m, \gamma) = \frac{\exp\left[-\frac{(\sigma-m)^2}{2\gamma^2}\right]}{\sqrt{2\pi}\gamma\left[1 - \Phi\left(-\frac{m}{\gamma}\right)\right]}, \tag{3}$$

where $\Phi(\cdot)$ denotes the cumulative distribution function of the standard normal distribution.

**Likelihood Function**. Let $\mathcal{T}$ denote the set of all forecast times selected for generating training data, and $\mathcal{I}$ denote the set of indices of time series, the $H$-horizon targets at time $t$ ($t \in \mathcal{T}$) of the $i$-th time series ($i \in \mathcal{I}$) is then $\mathbf{y}_{t+1:t+H}^i$, where $\mathbf{y}_{a:b}^i = (y_a^i, y_{a+1}^i, \ldots, y_b^i)^T$ denotes a slice of the $i$-th time series. Combining Equation equation 2 and equation 3, we can obtain the joint probability of the target set $\mathbf{Y} = \{\mathbf{y}_{t+1:t+H}^i : t \in \mathcal{T}, i \in \mathcal{I}\}$, which is

$$
\begin{aligned}
p(\mathbf{Y}; \Omega) &= \prod_{i \in \mathcal{I}} \prod_{t \in \mathcal{T}} p\left(\mathbf{y}_{t+1:t+H}^i | \sigma_t^i; \boldsymbol{\mu_t^i}, \kappa_t^i\right) p(\sigma_t^i; m_t^i, \gamma_t^i) \\
&= \prod_{i \in \mathcal{I}} \prod_{t \in \mathcal{T}} C_H(\kappa_t^i) \exp\left(\kappa_t^i (\boldsymbol{\mu_t^i})^T \times \frac{\mathbf{y}_{t+1:t+H}}{||\mathbf{y}_{t+1:t+H}||_2}\right) \frac{\exp\left\{-\frac{\left[||\mathbf{y}_{t+1:t+H}^i||_2 - m_t^i\right]^2}{2(\gamma_t^i)^2}\right\}}{\sqrt{2\pi}\gamma_t^i\left[1 - \Phi\left(-\frac{m_t^i}{\gamma_t^i}\right)\right]},
\end{aligned} \tag{4}
$$

where $\Omega = \{\{\boldsymbol{\mu_t^i}, \kappa_t^i, m_t^i, \gamma_t^i\} : t \in \mathcal{T}, i \in \mathcal{I}\}$, and $\sigma_t^i = ||\mathbf{y}_{t+1:t+H}^i||_2$.

---

[2] $I_n(z)$ is defined as $(z/2)^n \sum_{k=0}^{+\infty} \frac{(z/2)^{2k}}{k!\Gamma(n+k+1)}$, where $\Gamma(\cdot)$ is the Gamma function.

Let $\Theta$ denote the parameters of a neural network, and $\boldsymbol{\mu}, \kappa, m, \gamma$ be functions of the network output given $\Theta$, the history $\mathbf{y}_{0:t}^i$, and external features $\mathbf{x}^i$. Based on Equation equation 4, the likelihood function of the model parameters $\Theta$, is derived as follows:

$$\mathcal{L}(\Theta) = \prod_{i \in \mathcal{I}} \prod_{t \in \mathcal{T}} C_H(\kappa(\Theta, \mathbf{y}_{0:t}^i, \mathbf{x}^i)) \frac{\exp\left\{-\frac{\left[||\mathbf{y}_{t+1:t+H}^i||_2 - m(\Theta, \mathbf{y}_{0:t}^i, \mathbf{x}^i)\right]^2}{2\gamma^2(\Theta, \mathbf{y}_{0:t}^i, \mathbf{x}^i)}\right\}}{\sqrt{2\pi}\gamma(\Theta, \mathbf{y}_{0:t}^i, \mathbf{x}^i)\left\{1 - \Phi\left[-\frac{m(\Theta, \mathbf{y}_{0:t}^i, \mathbf{x}^i)}{\gamma(\Theta, \mathbf{y}_{0:t}^i, \mathbf{x}^i)}\right]\right\}}. \quad (5)$$

**Objective Function for MLE.** The log-likelihood function is obtained by taking the logarithm of Equation equation 6 as

$$l(\Theta) = C_l - \sum_{i \in \mathcal{I}} \sum_{t \in \mathcal{T}} \left[\left(\frac{H}{2} - 1\right) \log \kappa(\Theta, \mathbf{y}_{0:t}^i, \mathbf{x}^i) - \log I_{H/2-1}(\kappa(\Theta, \mathbf{y}_{0:t}^i, \mathbf{x}^i))\right] + \sum_{i \in \mathcal{I}} \sum_{t \in \mathcal{T}} \kappa(\Theta, \mathbf{y}_{0:t}^i, \mathbf{x}^i) \boldsymbol{\mu}^T(\Theta, \mathbf{y}_{0:t}^i, \mathbf{x}^i) \times \frac{\mathbf{y}_{t+1:t+H}^i}{||\mathbf{y}_{t+1:t+H}^i||_2}$$

$$- \sum_{i \in \mathcal{I}} \sum_{t \in \mathcal{T}} \frac{\left[||\mathbf{y}_{t+1:t+H}^i||_2 - m(\Theta, \mathbf{y}_{0:t}^i, \mathbf{x}^i)\right]^2}{2\gamma(\Theta, \mathbf{y}_{0:t}^i, \mathbf{x}^i)} - \sum_{i \in \mathcal{I}} \sum_{t \in \mathcal{T}} \left\{\log \gamma(\Theta, \mathbf{y}_{0:t}^i, \mathbf{x}^i) + \log\left[1 - \Phi\left(-\frac{m(\Theta, \mathbf{y}_{0:t}^i, \mathbf{x}^i)}{\gamma(\Theta, \mathbf{y}_{0:t}^i, \mathbf{x}^i)}\right)\right]\right\},$$

$$(6)$$

where $C_l$ represents the constant term. The training object is to maximize Equation equation 6 with respect to $\Theta$, or equivalently, minimize the inverse $-l(\Theta)$.

## 3.2 MODEL

**Transformer.** The seq2seq architecture is based on the Transformer (Li et al., 2020; Vaswani et al., 2017). The Transformer adopts an encoder-decoder structure and captures both long- and short-term dependencies via the multi-head self-attention mechanism.

**Model Output.** Let $\mathbf{h}$ denote the output of the decoder. We apply a fully-connected layer to $\mathbf{h}$ to obtain $\boldsymbol{\mu}$.

$$\boldsymbol{\mu} = \boldsymbol{W}_{\boldsymbol{\mu}}\mathbf{h} + \mathbf{b}_{\boldsymbol{\mu}}.$$

For $\kappa, m, \gamma$, we apply a fully-connected layer with soft-plus activation to ensure positivity. Specifically,

$$\kappa = \log(1 + \exp(\mathbf{w}_\kappa \mathbf{h} + b_\kappa)), m = \log(1 + \exp(\mathbf{w}_m \mathbf{h} + b_m)), \gamma = \log(1 + \exp(\mathbf{w}_\gamma \mathbf{h} + b_\gamma)).$$

**Self-Attention.** A self-attention function is to map a query and a set of key-value pairs to an output, where the query, keys, values, and output are all vectors (Vaswani et al., 2017). In order to compute the attention function on a set of queries simultaneously, all queries are usually packed into a query matrix $\boldsymbol{Q}$. The keys and values are also packed into matrices $\boldsymbol{K}$ and $\boldsymbol{V}$ respectively(Vaswani et al., 2017).

The conventional attention function measures the similarities between queries and keys, which are row vectors of $\boldsymbol{Q}$ and $\boldsymbol{K}$ respectively, by a 'dot-product' operation and feeds the similarities into a softmax function to normalize the similarities summing to 1 (Vaswani et al., 2017). Afterward, the output of the attention function is computed as the weighted average of the values, where the weights are the outputs of the softmax function. We adopt the multi-head convolutional self-attention mechanism, which has effectively enhanced awareness of local context, e.g. local shapes of time series(Li et al., 2020).

We modify the similarity measurement between queries and keys to better suit our objective function. We measure the similarity between a query vector and a key vector by computing the cosine of the angle between them and multiplying it with the difference between the length. For the rest of this article, we refer to this similarity measurement as the "Angle&Scale" similarity. The softmax function is then applied to obtain the weights of the values. Technically, for each time series $i$, let

$$\mathbf{z}_{t+1:t+H} = [y_t \circ \mathbf{x}_{t+1}^T \circ \ldots \circ \mathbf{x}_{t+H}^T]^T \in \mathbb{R}^{H*d+1}, \boldsymbol{Z}_t = [\mathbf{z}_{1:H}, \ldots, \mathbf{z}_{t+1:t+H}]^T \in \mathbb{R}^{(t+1) \times (H*d+1)}$$

where $[\cdot \circ \cdot]$ represents concatenation, and $\mathbf{x}_t \in \mathbb{R}^d$ represents the known inputs. The self-attention module first transforms $\boldsymbol{Z}_t$ into the query matrices $\boldsymbol{Q}$, key matrices $\boldsymbol{K}$, and value matrices $\boldsymbol{V}$ via

$$\boldsymbol{Q} = \boldsymbol{Z}_t * \boldsymbol{W}^Q = [\mathbf{q}_0, \ldots, \mathbf{q}_t]^T \in \mathbb{R}^{(t+1) \times d_k}, \boldsymbol{K} = \boldsymbol{Z}_t * \boldsymbol{W}^K = [\mathbf{k}_0, \ldots, \mathbf{k}_t]^T \in \mathbb{R}^{(t+1) \times d_k},$$

$$\boldsymbol{V} = \boldsymbol{Z}_t * \boldsymbol{W}^V = [\mathbf{v}_0, \ldots, \mathbf{v}_t]^T \in \mathbb{R}^{(t+1) \times d_v},$$

where $\boldsymbol{W}^K$, $\boldsymbol{W}^Q$ and $\boldsymbol{W}^V$ are learnable parameters, and the symbol $*$ represents the convolution operation (Li et al., 2020). The similarity between $\mathbf{q}_i$ and $\mathbf{k}_j$ is defined as

$$s_{i,j} = \frac{\mathbf{q}_i^T \mathbf{k}_j}{||\mathbf{q}_i||_2 ||\mathbf{k}_j||_2} \times \exp\left[-\left(||\mathbf{q}_i||_2 - ||\mathbf{k}_j||_2\right)^2\right]. \tag{7}$$

It should be noted that in Equation equation 7, the difference between $||\mathbf{q}_i||_2$ and $||\mathbf{k}_j||_2$ is passed into a Gaussian kernel with a radial parameter of 1. This operation is to ensure the difference ranging from 0 to 1, such that the magnitude of the cosine and difference would be at the same level.

Then, each row vector of the output matrices $\boldsymbol{O} = [\mathbf{o}_1, \ldots, \mathbf{o}_t]^T \in \mathbb{R}^{t \times (d_v + 1)}$, is defined as the normalized weighted average of the row vectors of the value matrices $\boldsymbol{V}$, concatenating its norm. By Equation equation 8, we formalize this definition as

$$\mathbf{o}_i = \left[\frac{\sum_{j=1}^t \mathrm{softmax}(s_{i,j})\mathbf{v}_j^T}{||\sum_{j=1}^t \mathrm{softmax}(s_{i,j})\mathbf{v}_j^T||_2} \circ ||\sum_{j=1}^t \mathrm{softmax}(s_{i,j})\mathbf{v}_j^T||_2\right]. \tag{8}$$

## 3.3 Training

The log-likelihood function is not directly differentiable because the Bessel function $I_d(x)$ cannot be written in a closed form (Kumar & Tsvetkov, 2018; Davidson et al., 2018). In addition, optimizing the Bessel function may cause an underflow problem when $d$ is large or $x$ is small (Kumar & Tsvetkov, 2018). Therefore, we alternatively optimize the upper bound of the logarithm of the Bessel function.

**Bound of Bessel Function**. We evaluate the lower and upper bounds of the logarithm of $I_d(x)$ and summarize the result in Proposition 3.1, and the provide the proof in Appendix. We also visualize the difference between the upper and lower bounds in Figure 4, which vividly illustrates the range of the approximation error.

**Proposition 3.1.** *Let $I_d(x)$ be the modified Bessel function of the first kind, and $m = d - \lfloor d \rfloor$, then*

$$\log\left(I_m(\kappa)\right) + \sum_{v=1}^{\lfloor d \rfloor} \log \frac{\kappa}{v + m - \frac{1}{2} + \sqrt{(v + m + \frac{1}{2})^2 + \kappa^2}} < \log\left(I_d(x)\right) < \log\left(I_m(\kappa)\right) + \sum_{v=1}^{\lfloor d \rfloor} \log \frac{\kappa}{v + m - 1 + \sqrt{(v + m + 1)^2 + \kappa^2}}$$

By Proposition 4, we use the following approximation of the logarithm of the Bessel function at the training stage,

$$\log\left(I_d(\kappa)\right) \sim \log\left(I_m(\kappa)\right) + \sum_{v=1}^{\lfloor d \rfloor} \log \frac{\kappa}{v + m - 1 + \sqrt{(v + m + 1)^2 + \kappa^2}}. \tag{9}$$

## 3.4 Prediction

At the prediction stage, there is no closed-form solution for computing the quantiles of $\mathbf{y}_{T+1:T+H}^i$. We alternatively draw random samples from the probability density function of $\mathbf{y}_{T+1:T+H}^i$ and estimate each quantile empirically.

**Random sampling**. We first sample the scale parameter $\sigma$ according to Equation equation 3 and next, we draw a sample of $\mathbf{y}_{T+1:T+H}^i$ according to Equation equation 2. Sampling $\mathbf{y}_{T+1:T+H}^i$ via Equation equation 2 is equivalent to first sampling a unit vector from a VMF distribution and multiplying it by $\sigma$. An $H$-dimensional random unit vector $\mathbf{y}$ subjecting to a VMF distribution with parameters $\boldsymbol{\mu}$, $\kappa$, can be decomposed as

$$\mathbf{y} = \boldsymbol{\mu} + \boldsymbol{v}\sqrt{(1 - t^2)}, \tag{10}$$

where $\boldsymbol{v}$ is a uniformly distributed unit tangent at $\boldsymbol{\mu}$, and $t \in [-1, 1]$ subjects to

$$p(t) = \frac{\left(\frac{\kappa}{2}\right)^{\frac{H}{2} - 1} \exp(\kappa t)(1 - t^2)^{\frac{H-3}{2}}}{\Gamma\left(\frac{H-1}{2}\right)\gamma\left(\frac{1}{2}\right)I_{\frac{H-1}{2}}(\kappa)} \tag{11}$$

Therefore, we respectively sample $t$ and $\boldsymbol{v}$, and construct a sample $\mathbf{y}$ by Equation equation 10. More details on the decomposition defined by Equation equation 10 can be referred to in (Mardia & Jupp, 2009).

## 4 EXPERIMENT

We analyze the approximation error of logarithm of Bessel function and evaluate the proposed methods on 3 datasets.

### 4.1 BESSEL FUNCTION APPROXIMATION ERROR

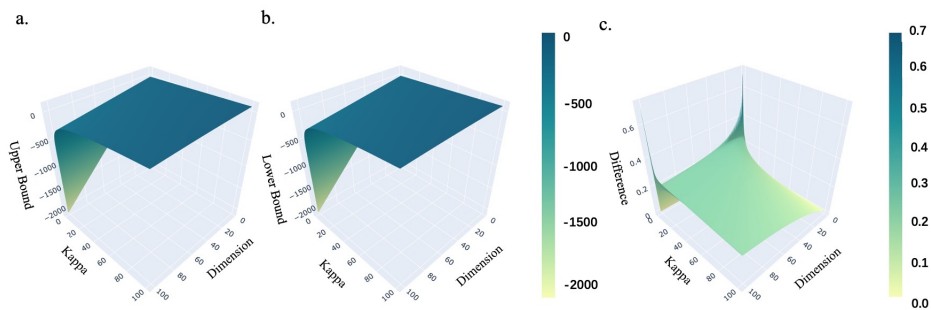

Figure 2: Upper (a) and lower (b) bounds of the logarithm of Bessel Function. The difference between the upper and lower bounds is shown in (c).

We evaluate the approximation error of the logarithm of the Bessel function given by Equation equation 9. We compute both the upper and lower bounds on a grid where $\kappa$ ranges from $1 \times 10^{-7}$ to 100, and $d$ ranges from 2 to 100. We plot the result in Figure 4. Besides, we observe that the logarithm of the Bessel function is bounded in a range of (-2093.07, 97.77) on the grid. The minimum and maximum values are achieved at $(\kappa, d) = (1 \times 10^{-10}, 100)$ and $(\kappa, d) = (100, 2)$ respectively. Recall that the underflow problem usually appears when $d$ is large (empirically larger than 5) and $\kappa$ is small; in our case, we avoid this problem via the approximation. Besides, the difference between the upper and lower bounds ranges from 0 to 0.72, and on about 95.7% of all grid points, the difference is smaller than 0.3. Since the absolute approximation error is smaller than the difference, we subsequently conclude that the upper bound of the absolute approximation error is smaller than 0.3.

Table 1: Comparison of $q$-risk of the VMFTransformer with competitive models on three public datasets. The prediction length (horizon) is set to H=24 and H=168. Best results are marked in bold (lower is better).

| H=24 | Electricity | | | Solar energy | | | Traffic | | |
|---|---|---|---|---|---|---|---|---|---|
| Model | q=50 | q=90 | AVG | q=50 | q=90 | AVG | q=50 | q=90 | AVG |
| DeepAR | 0.0734 | 0.0580 | 0.0657 | 0.4626 | 0.3097 | 0.3862 | 0.1526 | 0.1082 | 0.1304 |
| FeedForward | 0.0784 | 0.0433 | 0.0609 | 0.5341 | 0.3591 | 0.4466 | 0.2518 | 0.2011 | 0.2265 |
| TFT | 0.1107 | 0.0588 | 0.0848 | 0.4855 | 0.2471 | 0.3663 | 0.1981 | 0.1478 | 0.1730 |
| Transformer | 0.0789 | 0.0500 | 0.0645 | 0.5354 | 0.3502 | 0.4428 | 0.1644 | 0.1087 | 0.1366 |
| Informer | 0.1402 | 0.0729 | 0.1065 | 0.4722 | 0.2947 | 0.3834 | 0.6142 | 0.2947 | 0.4545 |
| Autoformer | 0.1224 | 0.0654 | 0.0939 | 0.2233 | 0.2113 | 0.2173 | 0.2157 | 0.1455 | 0.1806 |
| **VMFTransformer** | **0.0722** | **0.0427** | **0.0575** | **0.2125** | **0.1371** | **0.1748** | **0.1490** | **0.0935** | **0.1213** |
| H=168 | Electricity | | | Solar energy | | | Traffic | | |
| Model | q=50 | q=90 | AVG | q=50 | q=90 | AVG | q=50 | q=90 | AVG |
| DeepAR | 0.1464 | 0.0862 | 0.1163 | 0.5613 | 0.2349 | 0.3981 | 0.1553 | **0.1013** | 0.1283 |
| FeedForward | 0.1397 | 0.0730 | 0.1064 | 0.5891 | 0.2558 | 0.4224 | 0.2349 | 0.1360 | 0.1854 |
| Transformer | 0.1173 | 0.0670 | 0.0922 | 0.6319 | 0.3116 | 0.4717 | 0.1739 | 0.1063 | 0.1401 |
| TFT | 0.2243 | 0.1122 | 0.1682 | 0.6085 | 0.2030 | 0.4058 | 0.1514 | 0.1019 | 0.1267 |
| Informer | 0.1711 | 0.0749 | 0.1230 | 0.5434 | 0.2960 | 0.4197 | 0.7236 | 0.6800 | 0.7018 |
| Autoformer | 0.1342 | 0.0791 | 0.1066 | 0.2355 | 0.1589 | 0.1972 | 0.2559 | 0.1948 | 0.2254 |
| **VMFTransformer** | **0.0840** | **0.0598** | **0.0719** | **0.2318** | **0.1473** | **0.1896** | **0.1225** | 0.1099 | **0.1162** |

### 4.2 REAL-WORLD DATA EXPERIMENT

#### 4.2.1 DATASET.

We evaluate the performance of VMFTransformer on three public datasets, which are electricity, solar energy, and traffic. Electricity contains hourly time series of the electricity consumption of 370 customers ranging from 2012-01-01 to 2014-08-31 (Salinas et al., 2019). Solar energy, ranging

from 2006-01-01 to 2006-08-31, consists of 137 5-minute solar power time series obtained from the Monash Time Series Forecasting Repository (Godahewa et al., 2021). All solar power time series are aggregated to 1-hour granularity. Traffic, also used in (Salinas et al., 2019), contains the hourly measured occupancy rate, between 0 and 1, of 963 car lanes of San Francisco bay area freeways, ranging from 2008-01-02 to 2008-06-22. We follow the standard protocol and split all datasets into training, validation, and test sets in chronological order by the ratio of 7:1:2.

### 4.2.2 Implementation details and Baselines.

Our method is trained using the ADAM optimizer (Kingma & Ba, 2014) with an initial learning rate of $10^{-3}$. The batch size is set to 256. The training process is early stopped within five epochs. All experiments are implemented in PyTorch (Paszke et al., 2019).

We include a total number of 6 baseline methods. Specifically, we select four state-of- the-art transformer-based models: Informer (Zhou et al., 2021), Autoformer (Wu et al., 2021), Temporal Fusion Transformer (TFT) (Lim et al., 2021), the original Transformer (Vaswani et al., 2017)(Li et al., 2020), one RNN based method: DeepAR (Salinas et al., 2019), and one simple feed forward neural network.

### 4.2.3 Main Results.

We fix the input length as 168 and evaluate models with two prediction lengths: 24 and 168, corresponding to one-day and one-week horizons, respectively. We use three metrics, i.e., mean absolute error (MAE), mean squared error (MSE), and $q$-risk, to evaluate the performance of different methods. The first two measure the performance for forecasting the mean value, while the last quantifies the accuracy of a quantile $q$ of the predictive distribution (Salinas et al., 2019). We set $q = 50$ and 90 (Salinas et al., 2019).

We show in Table 1 and Table 2 that VMFTransformer achieves state-of-the-art performance. For $q$-risk, as directed in Table 1, VMFTransformer performs best in almost every benchmark except for one, which is q=90 and H=168 on the traffic dataset (Table 1). Still, the average $q$-risk is reduced (8.29%) by VMFTransformer relative to the second-best model (TFT). For MAE and MSE, we observe from Table 2 that VMFTransformer consistently outperforms all baselines.

Table 2: Comparison of MAE and MSE of the VMFTransformer with competitive models on three public datasets. The prediction length (horizon) is set to H=24 and H=168. The best results are marked in bold (lower is better).

| H=24 | Electricity | | | Solar energy | | | Traffic | | |
|---|---|---|---|---|---|---|---|---|---|
| Model | MAE | MSE | AVG | MAE | MSE | AVG | MAE | MSE | AVG |
| DeepAR | 0.0168 | 0.0188 | 0.0178 | 0.2619 | 0.3243 | 0.2931 | 0.1667 | 0.2042 | 0.1855 |
| FeedForward | 0.0153 | 0.0114 | 0.0133 | 0.3024 | 0.4213 | 0.3618 | 0.2751 | 0.3775 | 0.3263 |
| TFT | 0.0251 | 0.0363 | 0.0307 | 0.2749 | 0.3189 | 0.2969 | 0.2165 | 0.2912 | 0.2538 |
| Transformer | 0.0139 | 0.0144 | 0.0142 | 0.3031 | 0.4503 | 0.3767 | 0.1796 | 0.2050 | 0.1923 |
| Informer | 0.0845 | 0.0144 | 0.0494 | 0.2607 | 0.1824 | 0.2215 | 0.4620 | 0.6446 | 0.5533 |
| Autoformer | 0.1237 | 0.0253 | 0.0745 | 0.2106 | 0.1685 | 0.1895 | 0.2613 | 0.1542 | 0.2077 |
| **VMFTransformer** | **0.0116** | **0.0078** | **0.0097** | **0.2092** | **0.1334** | **0.1713** | **0.1334** | **0.1185** | **0.1259** |
| H=168 | Electricity | | | Solar energy | | | Traffic | | |
| Model | MAE | MSE | AVG | MAE | MSE | AVG | MAE | MSE | AVG |
| DeepAR | 0.0224 | 0.0301 | 0.0262 | 0.3077 | 0.3806 | 0.3442 | 0.1713 | 0.2115 | 0.1914 |
| FeedForward | 0.0211 | 0.0281 | 0.0246 | 0.3229 | 0.4122 | 0.3675 | 0.2590 | 0.2902 | 0.2746 |
| TFT | 0.0287 | 0.0469 | 0.0378 | 0.3336 | 0.5126 | 0.4231 | 0.1670 | 0.1952 | 0.1811 |
| Transformer | 0.0199 | 0.0296 | 0.0247 | 0.3464 | 0.4904 | 0.4184 | 0.1918 | 0.2081 | 0.2000 |
| Informer | 0.1461 | 0.0356 | 0.0908 | 0.2174 | 0.1562 | 0.1868 | 0.5591 | 0.8068 | 0.6829 |
| Autoformer | 0.1198 | 0.0242 | 0.0720 | 0.2785 | 0.1844 | 0.2314 | 0.2231 | 0.1747 | 0.1989 |
| **VMFTransformer** | **0.0128** | **0.0205** | **0.0167** | **0.2136** | **0.1390** | **0.1763** | **0.1390** | **0.1185** | **0.1288** |

### 4.2.4 Ablation Studies.

We use the solar energy dataset for the ablation study.
**Sensitivity to Sampling Size.** Since the prediction is conducted by random sampling, we study

Table 3: Comparison of the sensitivity of the VMFTransformer to the sampling size at the prediction step. The prediction length (horizon) is set to H=24 and H=168. Sample size (S) is set to 100, 1000, 10000, and 100000. The best results are marked in bold (lower is better). The stability of performance is measured by the average (AVG) divided by the standard deviation (STD).

| Samplesize | | 100 | 1000 | 10000 | 100000 | AVG | STD | STD/AVG |
|---|---|---|---|---|---|---|---|---|
| H=24 | MAE | 0.2341 | **0.2339** | 0.2424 | 0.2368 | 0.2368 | 0.0040 | 0.0168 |
| | MSE | 0.1846 | **0.1828** | 0.1995 | 0.1924 | 0.1898 | 0.0077 | 0.0406 |
| | 50-loss | 0.3243 | **0.3238** | 0.3370 | 0.3274 | 0.3281 | 0.0061 | 0.0187 |
| | 90-loss | **0.2435** | 0.2667 | 0.2490 | 0.2823 | 0.2604 | 0.0177 | 0.0678 |
| H=168 | MAE | **0.2000** | 0.2014 | 0.2005 | 0.2002 | 0.2005 | 0.0006 | 0.0030 |
| | MSE | 0.1303 | 0.1307 | **0.1286** | 0.1305 | 0.1300 | 0.0010 | 0.0077 |
| | 50-loss | **0.2830** | 0.2848 | 0.2834 | 0.2833 | 0.2836 | 0.0008 | 0.0028 |
| | 90-loss | 0.1733 | **0.1724** | 0.1749 | 0.1772 | 0.1744 | 0.0021 | 0.0118 |

Table 4: Comparison of similarity measurements of attention module. The best results are in bold (lower is better).

| Similarity | | Angle&Scale | Dot-product |
|---|---|---|---|
| H=24 | MAE | **0.2339** | 0.2389 |
| | MSE | 0.1828 | **0.1749** |
| | 50-loss | **0.3238** | 0.3317 |
| | 90-loss | **0.2667** | 0.2707 |
| H=168 | MAE | **0.2014** | 0.2082 |
| | MSE | **0.1307** | 0.1316 |
| | 50-loss | **0.2848** | 0.2950 |
| | 90-loss | **0.1724** | 0.2250 |

if the model performance is sensitive to sampling size. The sampling size (S) is set as 100, 1000, 10000, and 100000. In table 3, we show that the performance of VMFTransformer is relatively stable, especially when the prediction length is 168 (STD/AVG<1.2%). When the prediction length is short (24), S=1000 tends to show the optimal performance. Therefore, we recommend S=1000 for practical use.

**Attention Module.** We compare Angle&Scale similarity versus the original dot-product similarity for the attention module. We set the prediction length at H=24 and H=168. Table 4 shows that the Angle&Scale similarity outperforms the dot-product similarity in most cases (7 out of 8).

**Time Complexity.** Our Angle&Scale similarity requires more computation than the original dot-product similarity. The popular Transformer-based methods such as Reformer, Informer, and Autoformer have a theoretical time complexity of $O(L\log L)$, while the vanilla Transformer is $O(L^2)$, where $L$ is the encoding length. The theoretical time complexity of our VMFTransformer is still $O(L^2)$, which is the same as the vanilla Transformer.

## 5 CONCLUSION

We propose a probabilistic forecasting model termed VMFTransformer, which captures the temporal dependence of multi-horizon targets. Extensive experiments demonstrate that our model's performance is state-of-the-art on public datasets. The novel similarity measurement termed the "Angle&Scale" similarity is effective for multi-horizon time series forecasting.

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

# 6 APPENDIX

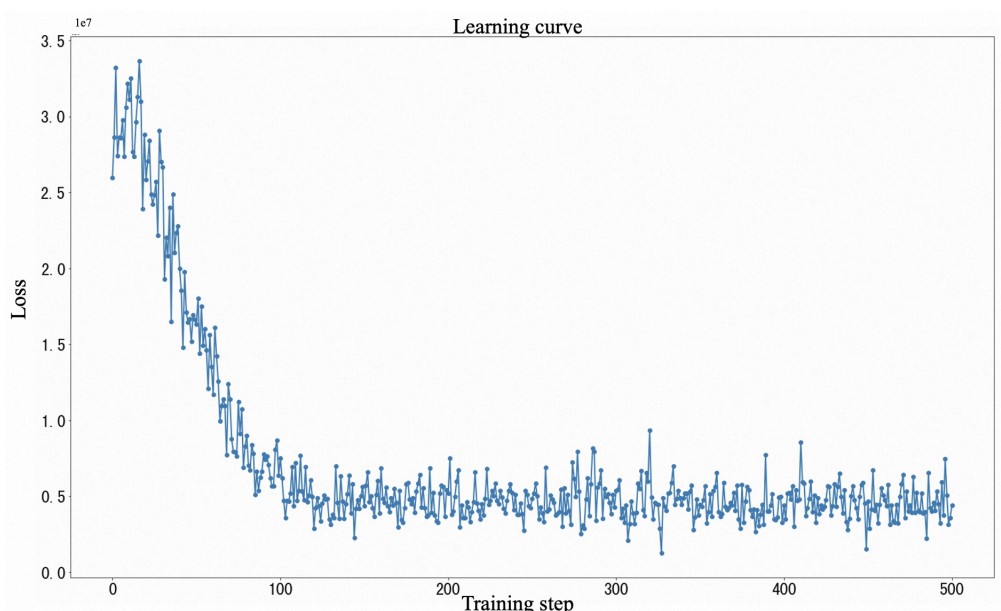

Figure 3: Learning curve of VMFTransformer on Solar, where the x-axis corresponds to learning steps and y-axis corresponds to training loss.

Table 5: Running time of different encoding length.

| Encoding length | Time (seconds) |
|---|---|
| 12 | 85.352 |
| 48 | 122.309 |
| 72 | 249.441 |
| 120 | 444.464 |
| 168 | 1629.578 |
| 240 | 2699.469 |
| 360 | 6767.158 |

Table 6: Memory usage of different encoding length.

| Encoding length | Memory (Byte) |
|---|---|
| 12 | 2498048 |
| 48 | 3061248 |
| 72 | 3462656 |
| 120 | 4314624 |
| 168 | 5248512 |
| 240 | 6776320 |
| 360 | 9700864 |

Table 7: Comparison of similarity measurements of attention module for both VMF-Loss and MSE-Loss. The best results are in bold (lower is better).

| VMF-Loss | | | |
|---|---|---|---|
| H | Similarity | Angle&Scale | Dot-product |
| 24 | MAE | **0.2339** | 0.2389 |
| | MSE | 0.1828 | **0.1749** |
| | 50-loss | **0.3238** | 0.3317 |
| | 90-loss | **0.2667** | 0.2707 |
| 168 | MAE | **0.2014** | 0.2389 |
| | MSE | **0.1307** | 0.1316 |
| | 50-loss | **0.2848** | 0.2950 |
| | 90-loss | **0.1724** | 0.2250 |
| MSE-Loss | | | |
| H | Similarity | Angle&Scale | Dot-product |
| 24 | MAE | 0.7389 | **0.7179** |
| | MSE | **0.4792** | 0.5061 |
| | 50-loss | 0.5059 | **0.5010** |
| | 90-loss | **0.8153** | 0.8650 |
| 168 | MAE | 0.7179 | 0.7179 |
| | MSE | 0.5075 | **0.5061** |
| | 50-loss | 0.5010 | 0.5010 |
| | 90-loss | **0.8631** | 0.8685 |

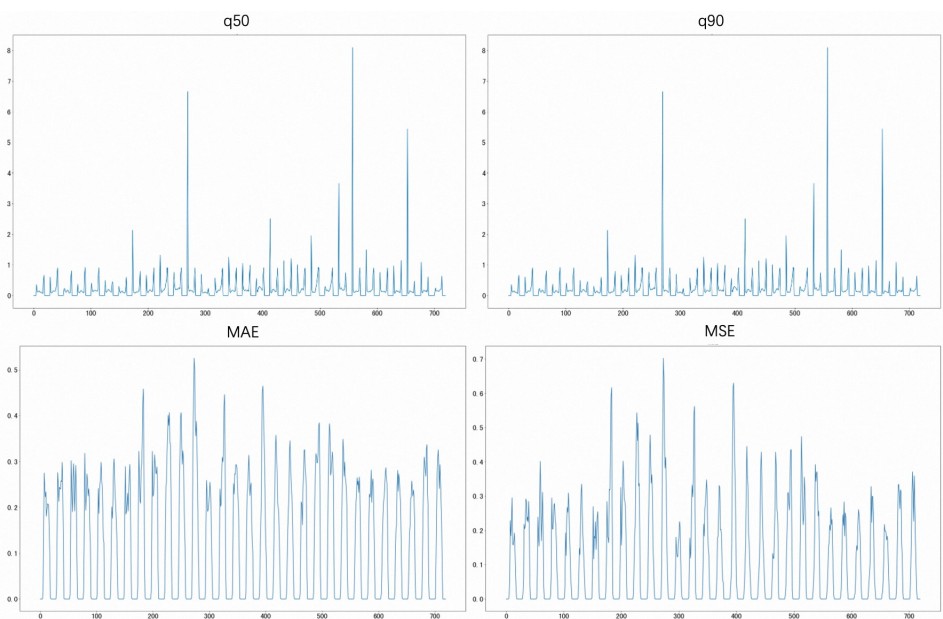

Figure 4: Comparison of metrics corresponding to different H (the x-axis) where H ranges from 1 to 720.

