# Appendix: VMFTransformer: An Angle-Preserving and Auto-Scaling Machine for Multi-horizon Probabilistic Forecasting

## Appendix

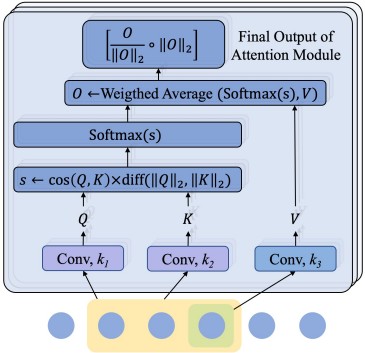

Figure 1: Attention mechanism. The input sequence is first passed into convolution kernels for computing queries ($Q$), keys ($K$), and values ($V$), where "Conv, $k_1$", "Conv, $k_2$", and "Conv, $k_3$" mean convolution of kernel size $k_1$, $k_2$, and $k_3$ with stride 1 respectively. The similarity between $Q$ and $K$ is measured by multiplying the cosine of the angle between them with the difference of their norms. The similarity is passed into a softmax function for computing the weighted average over $V$. The final output of the attention module is a concatenation of the direction of the weighted averaged $V$ (denoted by $O$) and its length, where the operator $[\cdot \circ \cdot]$ represents concatenation.

**Proposition .1.** *Let $I_d(x)$ be the modified Bessel function of the first kind, and $m = d - \lfloor d \rfloor$, then*

$$\log\left(I_m(\kappa)\right) + \sum_{v=1}^{\lfloor d \rfloor} \log \frac{\kappa}{v + m - \frac{1}{2} + \sqrt{(v + m + \frac{1}{2})^2 + \kappa^2}} < \log\left(I_d(x)\right) < \log\left(I_m(\kappa)\right) + \sum_{v=1}^{\lfloor d \rfloor} \log \frac{\kappa}{v + m - 1 + \sqrt{(v + m + 1)^2 + \kappa^2}}$$

*Proof.* Apparently,

$$\log\left(I_d(x)\right) = \log\left(I_m(x) \prod_{v=1}^{\lfloor d \rfloor} \frac{I_{m+v}(x)}{I_{m+v-1}(x)}\right) = \log\left(I_m(x)\right) + \sum_{v=1}^{\lfloor d \rfloor} \log\left(\frac{I_{m+v}(x)}{I_{m+v-1}(x)}\right).$$

By the Theorem 4 of Diego et al. 2016 Ruiz-Antolín & Segura (2016), for any $d \geq 0$

$$\frac{x}{d - \frac{1}{2} + \sqrt{(d + \frac{1}{2})^2 + x^2}} < \frac{I_d(x)}{I_{d-1}(x)} < \frac{x}{d - 1 + \sqrt{(d + 1)^2 + x^2}}.$$

We complete the proof. □

## References

Diego Ruiz-Antolín and Javier Segura. A new type of sharp bounds for ratios of modified bessel functions. *arXiv: Classical Analysis and ODEs*, 2016.