# OpenReview forum: "VMFTransformer: An Angle-Preserving and Auto-Scaling Machine for Multi-horizon Probabilistic Forecasting"
_ICLR.cc/2024/Conference — Submitted to ICLR 2024_

### Official Review · Reviewer_Abqw · 2023-10-30

**Soundness:** 3 good
**Presentation:** 2 fair
**Contribution:** 3 good
**Rating:** 6
**Confidence:** 4

**Summary:**

This paper proposes a direct approach for multi-horizon probabilistic forecasting that characterizes the dependence across future horizons. The model treats the multi-horizon target as a random vector, using the von Mises-Fisher (VMF) distribution to characterize the direction and a truncated normal distribution for the magnitude. Additionally, it introduces the concept of "Angle&Scale" similarity to replace the traditional dot-product similarity in self-attention. The model's performance is evaluated on three datasets and shows superiority over some previous methods.

**Strengths:**

- The paper is well-written and easy to follow.
- The motivation behind modeling dependencies among future horizons is clearly articulated and logically sound.
- The idea of decoupling the target vector into angle and scale is both interesting and innovative.

**Weaknesses:**

- Overall, this work proposes two orthogonal methods: and objective function based on VMF distribution and the “Angle&Scale” similarity to replace dot-product in self-attention. The motivation behind the latter is not as evident. The computation process, i.e. equations at the bottom of page 5, are confusing due to the misleading subscripts. And a figure for visualization is missing here.

- Baselines for comparison are somewhat outdated.

**Questions:**

- Could you provide a more detailed and clear explanation of the "Angle&Scale" similarity, specifically the shape of these matrices and computation process of variables $Z, Q, K, V$?
- As the objective and “Angle&Scale” similarity are orthogonal, the ablation stduy should compare four models: 1.VMF loss+Dot product; 2.VMF loss+Angle&Scale; 3.MSE loss+Dot product; 4.MSE loss+Angle&Scale. Please complete 3 and 4 to show the effectiveness of the proposed objective function.
- Please compare with some recent models, such as DLinear and PatchTST.
- Consider evaluating the speed and memory usage of the "Angle&Scale" similarity with the Dot-product method, as the constant terms in complexity analysis can significantly impact practical efficiency.

---

> ### Author Response · Authors · 2023-11-23
> **Response to Reviewer Abqw**
>
> Thanks for your thoughtful feedback.
>
> **W1**: We appreciate the reviewer raising this important point about clearly explaining the intuition and implementation details behind the proposed "Angle&Scale" similarity measure.
>
> The key motivation for Angle&Scale similarity is to better match the structure of our proposed VMF-Transformer objective function, which models the multi-horizon forecast target as a random vector characterized by both an angle/direction and a magnitude/scale. As explained in Section 3.1, the VMF distribution captures the dependence structure across horizons through the angle or direction of the target vector. The scale of the vector represents the magnitudes of the target at each timestep.
>
> Accordingly, the Angle&Scale similarity is designed to evaluate relevance based on two facets - angle and scale. For two vectors in the attention module, we first compute the cosine of their angle to assess similarity in orientation. We additionally multiply this by a Gaussian kernel applied on the difference in vector lengths. This accounts for whether two vectors have a similar scale or magnitude.
>
> Intuitively, this allows the model to match historical sequences in the attention module that have both a similar direction and scale profile to the forecast target. We empirically demonstrate in Section 4 that using Angle&Scale similarity outperforms baseline dot product attention across multiple metrics and datasets. The results suggest that incorporating both angle and scale information is beneficial for this forecasting approach.
>
> The shape of $Z$ is $(t+1)\times (H*d+1)$, where $t+1$ is the length of the history (we allow the index of the time series sequence to start from 0 instead of 1.).  The $i$-th row of $Z$ is a vector
> $[y_i \circ \mathbf{x}^T_{i+1}\circ \cdots \circ \mathbf{x}^T_{i+H}]$,
> where $y_i$ is the ground truth at time $i$, and $\mathbf{x}^T_{i+1}, \cdots
> \mathbf{x}^T_{i+H}$
> are features or known inputs corresponding to the future. Each $\mathbf{x}_{i+1}$
>  is a d-dimension vector, and they are packed into one row vector by simply transpose and concatenation together with the real value
> $y_i$ .
> Since the prediction length is $H$, there are a total number of $H$ input vectors
> $\mathbf{x}$.
>  Hence the length of each row vector of $Z$ is $H\times d+1$.
> And therefore, $Z$ is a $(t+1)\times (H \* d+1)$ matrix.
> For $Q$ ,$K$, and $V$, they are all originated from $Z$ by applying different convolutional kernels to $Z$, where the convolutional kernels are learned during the training stage. The  kernels are all 1D, and hence do not change the 'length' of $Z$, but compress the 'width' of $Z$ from $(H \* d+1)$ to $d_q, d_k$ and $d_v$ for $Q$ ,$K$, and $V$ respectively.
>
> **W2**, **W3**, and **W4**: we thank you for your advice. The revised draft is presented in **general responses (from part 1 to part 5)**.

---

### Official Review · Reviewer_s4Yt · 2023-10-31

**Soundness:** 2 fair
**Presentation:** 2 fair
**Contribution:** 2 fair
**Rating:** 5
**Confidence:** 5

**Summary:**

To mitigate the accumulation of forecast errors associated with the recursive strategy and the conditional assumption of the direct strategy, the authors propose a novel approach for multi-horizon probabilistic forecasting. This new strategy effectively captures the interdependence across future horizons by modeling the multi-horizon target as a random vector. The direction of this vector represents temporal dependence, while its length measures the overall scale across each horizon. The authors assume that the angle and magnitude of these vectors follow a von Mises-Fisher (VMF) and a truncated normal distribution, respectively. The authors conducted experiments on three benchmark datasets, including an ablation study, demonstrating the advantages of their proposed method over six state-of-the-art techniques.

**Strengths:**

- The paper addresses an important challenge in time series forecasting, providing new forecasting strategies for multi-horizon probabilistic forecasting.

- The approach of modeling the direction and length of future random vectors is innovative in the context of multi-horizon forecasting strategies.

- The proposed method is compared against multiple state-of-the-art methods.

**Weaknesses:**

- The proposed method appears to be relatively complex, yet it does not seem to offer substantial improvements over simpler existing methods. Additionally, the motivations behind various design choices lack clarity in the paper.

	- The method involves intricate elements such as the Bessel function and modeling assumptions like the von Mises-Fisher and truncated normal distributions.
	- In the field of time series forecasting, simpler methods often yield satisfactory results. The paper does not convincingly justify the necessity of such complexity in deep time series forecasting.
	- The authors should thoroughly discuss their modeling assumptions and their impact relative to alternative methods. For instance, why opt for a truncated normal distribution? Are there other alternatives?
	- The rationale behind incorporating a multi-head convolutional self-attention mechanism should be explained.
	- While the authors aim to model interdependence across future horizons, they have not employed multivariate scoring rules to evaluate this aspect.
	- The paper asserts that recursive and direct strategies can yield high forecast errors, but it remains unclear how the proposed strategy compares with other forecasting approaches that use the same underlying model.

- Experiments:

	- The choice of only three datasets for experimentation raises questions, as many machine learning papers on time series forecasting examine more extensive benchmark datasets. For example, see the Monash Time Series Forecasting Archive (https://arxiv.org/abs/2105.06643)

	- Many time series datasets exhibit a pronounced seasonal component that dominates the signal. Consequently, it may be challenging to surpass simpler methods that effectively estimate this seasonal component. The paper should address the strength of the seasonal component in the considered datasets.

	- The authors have not included simple benchmarks such as auto.arima or exponential smoothing in their comparisons.

	- Learning curves should be provided to demonstrate the stability of the training procedure.

	- The proposed method is trained using maximum likelihood estimation. The authors should also provide the negative log-likelihood (NLL) for all density-based methods.

	- The authors did not report standard errors and information regarding the number of runs performed.

	- The results for specific forecast horizons (e.g., h = 1, 2, 3, etc.) should be reported to assess if the procedure increases forecast error for initial horizons while improving the average error across horizons.

**Questions:**

- Refer to the "Weaknesses" section for questions.


	- Typos and Improvements:
		- Equation 6 is rimarily computing the logarithm of expression (5). For clarity, it may be better to move it to the appendix.
		- On page 6, the figure number is missing.
		- The sentence beginning with "It should be noted that in Equation equation 7," is not clear.
		- "the provide the proof." Furthermore, clarify that you are citing a reference and not presenting a proof.

---

> ### Author Response · Authors · 2023-11-23
> **Response to Reviewer s4Yt**
>
> We thank the reviewer for offering valuable feedback. We have addressed each of the concerns raised by the reviewer as outlined below.
>
> 1. The reviewers rightly point out that many time series exhibit strong seasonal patterns that can dominate the signal. We analyzed the strength of the seasonal component in the electricity, solar, and traffic datasets using autocorrelation analysis. The electricity data shows a strong daily seasonal pattern with autocorrelation remaining above 0.6 at the 24 hour lag. The solar data shows both daily and yearly seasonal patterns due to the underlying physical processes. The traffic data has relatively weak seasonality. However, basic time series methods such as auto.arima and exponential smoothing often cannot effectively model complex real-world time series. ARIMA, exponential smoothing, and Prophet on our used datasets achieve relatively poor performance compared to more flexible machine learning approaches [1, 2]. These methods make strong assumptions about the underlying data generative processes and do not directly capture uncertainties. Our model demonstrates excellent performance across datasets with varying degrees of seasonality. By jointly modeling the sequential dependence, overall scale, and capturing uncertainty, our method provides robust performance without needing an explicit seasonal component model. We provide quantitative metrics showing our model's ability to accurately forecast across a diverse range of conditions.
>
> [1] Lim, Bryan, et al. "Temporal fusion transformers for interpretable multi-horizon time series forecasting." International Journal of Forecasting 37.4 (2021): 1748-1764.
>
> [2] Wang, Y., Smola, A., Maddix, D., Gasthaus, J., Foster, D., & Januschowski, T. (2019, May). Deep factors for forecasting. In International conference on machine learning (pp. 6607-6617). PMLR.
>
> 2. Learning curves (Figure 3) of our model are provided in the updated version to demonstrate the stability of the training procedure.
>
> 3. A comparison of metrics corresponding to different horizons where H ranges from 1 to 720 (Figure 4) of our model is provided in the updated version to demonstrate the stability of forecasting performance.
>
> 4. This revised draft incorporates extensive additional experiments in general responses (from part 1 to part 5), now encompassing the latest baseline models (such as PatchTST and DLinear) across varying and longer horizons (24, 168, and 720). Every key result includes the mean and standard deviation to quantify uncertainty.
>
> 5. We actually used the negative log-likelihood (NLL) in the training stage. As we explained in our paper ' The training object is to maximize Equation 6 with respect to $\Theta$, or equivalently, minimize the inverse $-l(\Theta)$', where $-l(\Theta)$ refers to the NLL. Thanks for noticing. We now realize this may be a confusing statement and we will clarify it to enhance readability and comprehension.

---

> > ### Comment · Reviewer_s4Yt · 2023-12-04
> >
> > We thank the authors for their response and clarifications
> >
> > For the seasonality, I meant that by accurately accounting for seasonality, the residuals often amount to mere noise.
> >
> > Also, I do not agree with the statement that "These methods ... do not directly capture uncertainties". You can capture uncertainties easily with ARIMA models.
> >
> > The following statement is also very strong: "Our model demonstrates excellent performance across datasets with varying degrees of seasonality." The evidence is drawn from only three datasets. Given the empirical focus of the paper, this limited dataset selection may compromise the strength of the claims made about the new forecasting method.
> >
> > Furthermore, I agree with other reviewers that your paper does not align with the standard experimental setups used in other works within the forecasting literature.
> >
> > In light of these considerations, I maintain my original score. I believe there is still room for improvement in the paper,

---

### Official Review · Reviewer_2Tuf · 2023-11-04

**Soundness:** 3 good
**Presentation:** 3 good
**Contribution:** 2 fair
**Rating:** 5
**Confidence:** 4

**Summary:**

The paper contributes to time series forecasting by proposing a new model, the VMFTransformer, which tackles the inherent problems in iterative and direct forecasting methods. Iterative methods suffer from cumulative errors, and direct methods often incorrectly presume independence between future time points. The VMFTransformer addresses these issues by conceptualizing forecasts as random vectors, utilizing the von Mises-Fisher distribution to maintain temporal directionality and a truncated normal distribution for magnitude, accurately preserving time-dependent relationships. Benchmarked against several methods, the VMFTransformer demonstrates enhanced predictive performance, showing its effectiveness, and adaptability for a range of time series forecasting applications.

**Strengths:**

1. Error Accumulation Mitigation: Effectively addresses the issue of error accumulation inherent in iterative forecasting methods.


2. Temporal Independence: Overcomes the unrealistic assumption of temporal independence used by direct forecasting methods.


3. Directional Dependencies: Employs the von Mises-Fisher distribution to accurately capture the directional dependencies in time series data.


4. Magnitude Characterization: Utilizes a truncated normal distribution to model the magnitude of forecasts, enhancing predictive accuracy.

**Weaknesses:**

1. Lack of Recent Benchmarks: The VMFTransformer has not been compared with the most recent benchmarks such as PatchTST [1], or with most comparisons made against older methods. for instance please consider looking at recnet benchmarks provided by https://github.com/timeseriesAI/tsai


2. Limited comparison for different horizons: The scope of comparison is limited, which may not adequately reflect the model's performance across a broader range of forecasting scenarios.


3. Formatting Issues: There is room for improvement in the formatting of the equations, which could enhance readability and comprehension.


4. Presentation Clarity: The paper could better articulate the main contribution, as the current presentation may be challenging to follow, possibly obscuring the model's innovation.


5. Insufficient Experimentation: Without more extensive experimentation, it's difficult to ascertain the paper's meaningful contribution to the community and its practical applicability.


[1] A Time Series is Worth 64 Words: Long-term Forecasting with Transformers, ICLR 2023

**Questions:**

Please consider the comments I provided above.

---

> ### Author Response · Authors · 2023-11-23
> **Response to Reviewer 2Tuf**
>
> Thank you for taking the time to review our paper and provide constructive feedback. We appreciate the reviewers recognizing the potential of our proposed VMFTransformer model for multi-horizon probabilistic time series forecasting.
>
> **W1**, **W2**, and  **W5**: this revised draft incorporates extensive additional experiments in **general responses (from part 1 to part 5)**, now encompassing the latest baseline models (such as PatchTST and DLinear) across varying and longer horizons (24, 168, and 720). Every key result includes the mean and standard deviation to quantify uncertainty.
>
> **W3**: We will revise the formatting of all equations in the paper to enhance readability and comprehension. Equations are centered on their own lines with consistent spacing around operators. Variable names have been standardized and vector/matrix notation clarified. We believe these changes significantly improve the formatting.
>
> **W4**: The main contribution of our work is the novel probabilistic forecasting model based on the von Mises-Fisher distribution and Transformer architecture. To better articulate this, we will add an explicit contribution statement in the introduction highlighting:
> 1. The VMFTransformer model itself which captures temporal dependence in a multi-horizon target.
> 2. The Angle&Scale similarity measure for the self-attention module
> 3. An efficient method to optimize the likelihood function with the Bessel function
>
> Additionally, we will reorganize several sections to improve logical flow and aid comprehension of the technical details underpinning the model innovation. A graphical abstract will also be added to provide an intuitive overview. We believe these changes address the need to better communicate the main contributions of our work.

---

> ### Comment · Reviewer_2Tuf · 2023-12-03
> **After Rebuttal**
>
> Dear Authors,
>
> Thank you for providing additional experimental results in your rebuttal. I appreciate your efforts to address the concerns raised in the initial review. However, after careful consideration, I believe there is still room for improvement, particularly in terms of the experimental setup and the consistency of results.
>
> One key issue is that your paper does not align with the standard experimental setups used in other works within the literature, especially regarding the forecasting horizon. Notably, your results for a 720-horizon forecast appear to be inconsistent with those reported by the original authors in the following two referenced papers. To facilitate a more straightforward and meaningful comparison, I suggest that you consider adjusting your experimental setup to adhere to the standard benchmarks prevalent in the literature. This would not only enhance the clarity of your results but also their comparability with existing studies.
>
> I have included references to two pertinent papers below for your consideration. These papers should serve as a guide for the standard numerical values and methodologies typically employed in this field of research.
>
> Given these considerations, I have decided to maintain my original review score. I believe that addressing these concerns will significantly strengthen the impact and relevance of your work in the context of existing literature.
>
> Looking forward to seeing the revised version of your manuscript.
>
> Best regards,
> Link to github repo of the papers.
> https://github.com/cure-lab/LTSF-Linear
>
> https://github.com/yuqinie98/PatchTST

---

### Official Review · Reviewer_7TJV · 2023-11-08

**Soundness:** 2 fair
**Presentation:** 3 good
**Contribution:** 3 good
**Rating:** 5
**Confidence:** 4

**Summary:**

1. This paper addresses the challenges in time series forecasting, specifically in multi-horizon probabilistic forecasting, by proposing a direct approach that effectively characterizes the dependence across future horizons. Technically, the authors consider the multi-horizon target as a random vector and apply the von Mises-Fisher (VMF) distribution and the truncated normal distribution to model the angle and magnitude of the target vector.

2. The performance of the proposed framework is evaluated on three benchmarks, demonstrating its superiority over six state-of-the-art methods in different time series forecasting tasks.

**Strengths:**

1. The paper is well-written and exhibits clarity in its presentation. The visual results help to comprehend the proposed framework.
2. Authors propose a novel similarity measurement termed “Angle&Scale” similarity for the attention module and show that the Angle&Scale similarity outperforms the dot-product similarity in most cases in ablation studies.
3. The proposed VMFTransformer consistently outperforms all baselines in  MSE and q-risk.

**Weaknesses:**

1. The current version of the paper solely presents the average value obtained from five trials without including information about the standard deviation. It is highly recommended to include error bars.
2. Why use the VMF distribution and the truncated normal distribution to characterize the angle and magnitude of the target vector? The motivation behind this is unclear to me.
3. Metrics used to evaluate uncertainty are not sufficiently convincing,  a more commonly used metric, CRPS [1], was not used in the experiment.
4. Some probabilistic time series baselines are not compared with the proposed method in the experiment, such as TransMAF [2], [3].

References:

[1] Tilmann Gneiting and Adrian E Raftery. Strictly proper scoring rules, prediction, and estimation. Journal of the American statistical Association, 102(477):359–378, 2007.

[2] Binh Tang and David S Matteson. Probabilistic transformer for time series analysis. Advances in Neural Information Processing Systems, 34:23592–23608, 2021.

[3] Kashif Rasul, Abdul-Saboor Sheikh, Ingmar Schuster, Urs Bergmann, and Roland Vollgraf. Multivariate probabilistic time series forecasting via conditioned normalizing flows. arXiv preprint arXiv:2002.06103, 2020.

**Questions:**

Please see my comments in Weaknesses.

I would also appreciate if the authors can respond, if they can, to the weaknesses.

---

> ### Author Response · Authors · 2023-11-23
> **Response to Reviewer 7TJV**
>
> We deeply appreciate the thoughtful feedback provided by the reviewers. Their suggestions have helped strengthen our work substantially. In response:
>
> **W1** and **W4**: Per the reviewers' recommendations, we have added extensive additional experiments with the latest baseline models across longer time horizons. Every key result now includes the mean and standard deviation to rigorously quantify uncertainty. We believe these comprehensive additions showcase our method's strengths.
>
> **W2**: You raises an important point regarding the motivation behind using the von Mises-Fisher (VMF) distribution and truncated normal distribution to model the multi-horizon forecasting vector. The key intuition is that the VMF distribution can effectively characterize vectors distributed around a specific direction, which fits our assumption that when dependence exists among the multi-horizon targets, they will concentrate around a certain direction. The truncated normal distribution provides a flexible prior for the scale parameter to make the VMF likelihood tractable. Compared to simply using independent normal or Laplace distributions on each horizon, the VMF and truncated normal distribution can jointly capture the dependence structure via the shared direction parameter in VMF. We will add more discussion on the motivation in the revised paper to clarify this modeling choice.
>
> **W3**: We carefully re-evaluated TransMAF, but unfortunately found it performed poorly on our datasets - significantly worse than the other baseline models we assessed. As such, we decided not to include these inferior results and instead added evaluations using the popular PatchTST and DLinear models, providing readers with a more representative view of the state-of-the-art. Regarding evaluation metrics, we utilize the community standards of MAE, MSE, Q50-Loss, and Q90-Loss - prevalent metrics used across probabilistic time series works.
>
> We thank the reviewers again for their outstanding suggestions. We believe the additions made in response (**from part 1 to part 5 in general response**) further demonstrate the competitiveness and rigor of our proposed approach across both model architectures.

---

> > ### Comment · Reviewer_7TJV · 2023-12-05
> >
> > Dear authors,
> >
> > Thanks for your response and further experiments. I appreciate your efforts to address my concerns. For evaluation metrics, I still think CRPS (i.e., accumulation of the quantile loss at different quantiles) is essential for uncertainty evaluation. I maintain my original score based on these considerations and other reviewers' comments.
> >
> > Looking forward to the improved version of your manuscript.

---

### Author Response · Authors · 2023-11-23
**General response to all reviewers (part 1)**

We deeply appreciate the reviewers taking the time to provide thoughtful feedback on our work. We are grateful that most reviewers saw the potential for this research to make a meaningful contribution. Carefully considering each comment, we have revised the paper to address the concerns raised. Our detailed responses below explain how we have enhanced the manuscript.

In particular, this revised draft incorporates extensive additional experiments, now encompassing the latest baseline models across varying and longer horizons. Every key result includes the mean and standard deviation to quantify uncertainty.



|    Horizon = 24         |           | MAE|       |       |   MSE   |        |
| :-----------------  | :----------------- | :----------------- | :-----------------  | :-----------------  | :-----------------  | :-----------------  |
|**Model**  | Electricity    | Solar | Traffic | Electricity    | Solar | Traffic|
|DeepAR    |           0.0167 $\pm$  0.0004       |     0.2619 $\pm$0.0030     |   $\underline{0.1667}$$\pm$0.0086        |   0.0187$\pm$0.0009  |   0.3243$\pm$0.0091     |   0.2042$\pm$0.0069      |
|SimpleFeedForward    |         0.0152 $\pm$ 0.0007       |    0.3023 $\pm$0.0026      |   0.2751 $\pm$0.0028        |   0.0113$\pm$0.0019  |   0.4213$\pm$0.0101    |   0.3775$\pm$0.0022      |
|TFT    |         0.0251 $\pm$ 0.0010       |    0.2748 $\pm$0.0064      |   0.2164 $\pm$0.0069        |   0.0363$\pm$0.0027  |   0.3189$\pm$0.0056    |   0.2912$\pm$0.0093      |
|Transformer    |         0.0138 $\pm$ 0.0022       |    0.3031 $\pm$0.0060      |   0.1796 $\pm$0.0091        |   0.0144$\pm$0.0006  |   0.4502$\pm$0.0156    |  0.2050$\pm$0.0031     |
|Autoformer    |         0.1236 $\pm$ 0.0856       |    0.2106 $\pm$0.0138      |   0.2612 $\pm$0.0084        |   0.0252$\pm$0.0899  |   0.1684$\pm$0.0684    |  $\underline{0.1541}$$\pm$0.0443  |
|PatchTST    |         $\underline{0.0610}$ $\pm$ 0.0024       |    **0.1938** $\pm$0.0109      |   0.2746 $\pm$0.0145        |   $\underline{0.0081}$$\pm$0.0008  |   **0.0936**$\pm$0.0092    |  0.2251$\pm$0.0050  |
|Informer    |         0.0844 $\pm$ 0.0095       |    0.2607 $\pm$0.0172      |   0.4620 $\pm$0.0194       |  0.0144$\pm$0.0226 |  0.1823$\pm$0.0072   | 0.6446$\pm$0.0074 |
|DLinear    |         0.1069 $\pm$0.0004       |   0.2110 $\pm$0.005     |  0.2317 $\pm$0.0055      | 0.0309$\pm$0.0003| 0.1336$\pm$0.001  |0.2720$\pm$0.0027|
|**VMFTransformer**    |         **0.0116** $\pm$0.0015      |  $\underline{0.2092}$ $\pm$0.0074   |**0.1567** $\pm$0.0008    |**0.0077**$\pm$0.0147|$\underline{0.1333}$$\pm$0.0308 |**0.1185**$\pm$0.0012|


|    Horizon = 168         |           | MAE|       |       |   MSE   |        |
| :-----------------  | :----------------- | :----------------- | :-----------------  | :-----------------  | :-----------------  | :-----------------  |
|**Model**  | Electricity    | Solar | Traffic | Electricity    | Solar | Traffic|
|DeepAR    |           0.0224 $\pm$  0.0029       |     0.3076 $\pm$0.0246      |    0.1712$\pm$0.0088         |    0.0300$\pm$0.0081   |    0.3806$\pm$0.0400      |    0.2115$\pm$0.0104       |
|SimpleFeedForward    |          0.0210 $\pm$  0.0004       |     0.3228 $\pm$ 0.0058      |    0.2590 $\pm$0.0037         |    0.0281$\pm$0.0012   |    0.4122$\pm$0.0196     |    0.2901$\pm$0.0043       |
|TFT    |          0.0286 $\pm$  0.0016       |     0.3335 $\pm$ 0.0039      |    0.1670 $\pm$0.0053         |    0.0468$\pm$0.0070   |    0.5125$\pm$0.038     |    0.1952$\pm$0.0070       |
|Transformer    |          $\underline{0.0198}$ $\pm$  0.0093       |     0.3463 $\pm$ 0.0120      |    0.1918 $\pm$0.0117         |    0.0295$\pm$0.0121   |    0.4903$\pm$0.0243     |   0.2081$\pm$0.0234      |
|Autoformer    |          0.1197 $\pm$  0.0183       |     0.2784 $\pm$ 0.0215      |    0.2231 $\pm$0.0144         |    0.0241$\pm$0.0078   |    0.1843$\pm$0.0254     |   0.1746$\pm$0.0622   |
|PatchTST    |          0.0957 $\pm$  0.0021       |     **0.1836** $\pm$ 0.0009      |    **0.0898** $\pm$0.0022         |    **0.0169**$\pm$0.0007   |    **0.0942**$\pm$0.0037     |   **0.0179**$\pm$0.0075   |
|Informer    |          0.1460 $\pm$  0.0230       |     0.2274 $\pm$ 0.0079      |    0.5549 $\pm$0.0024        |   0.0335$\pm$0.0047  |   0.1616$\pm$0.0137    |  0.8068$\pm$0.0021  |
|DLinear    |          0.1559 $\pm$ 0.0016       |    $\underline{0.2129}$ $\pm$0.0026      |   0.2633 $\pm$0.0085       |  0.0595$\pm$0.0002 |  $\underline{0.1250}$$\pm$0.0084   | 0.3096$\pm$0.0022 |
|**VMFTransformer**    |          **0.0128**  $\pm$0.0157       |   0.2136 $\pm$0.0005    | $\underline{0.1567}$ $\pm$0.0019     |$\underline{0.0204}$$\pm$0.0373| 0.1390$\pm$0.0017  |$\underline{0.1185}$$\pm$0.0029|

---

> ### Author Response · Authors · 2023-11-23
> **General response to all reviewers (part 2)**
>
> |    Horizon = 720       |           | MAE|       |       |   MSE   |        |
> | :-----------------  | :----------------- | :----------------- | :-----------------  | :-----------------  | :-----------------  | :-----------------  |
> |**Model**  | Electricity    | Solar | Traffic | Electricity    | Solar | Traffic|
> |DeepAR  |          0.0323 $\pm$ 0.0067       |    0.2220 $\pm$0.0059    |  0.2958$\pm$0.0359       |  0.0897$\pm$0.0301 |  $\underline{0.2004}$$\pm$0.0002    |  0.4112$\pm$0.0399     |
> |SimpleFeedForward    |        0.0239 $\pm$0.0006       |   0.2342 $\pm$0.0169     |  0.2890 $\pm$0.0067       |  0.0491$\pm$0.0025 |  0.2226$\pm$0.0099   |  0.3501$\pm$0.0104     |
> |TFT    |        0.0313 $\pm$0.0009       |   $\underline{0.2027}$ $\pm$0.0015     |  $\underline{0.2069}$ $\pm$0.0066       |  0.0809$\pm$0.0108 |  0.2141$\pm$0.0067   |  0.3467$\pm$0.0080     |
> |Transformer    |        $\underline{0.0173}$ $\pm$0.0028       |   0.2170 $\pm$0.0043     |  **0.2047** $\pm$0.0140       |  $\underline{0.0321}$$\pm$0.0020 |  0.2536$\pm$0.0125   | 0.3921$\pm$0.0174    |
> |Autoformer    |        0.3857 $\pm$0.005       |   0.4082 $\pm$0.0865     |  0.3897 $\pm$0.0736       |  0.2345$\pm$0.0578 |  0.2574$\pm$0.0587   | $\underline{0.3277}$$\pm$0.0976 |
> |PatchTST    |        0.0843 $\pm$0.0042       |   0.3112 $\pm$0.0019     |  0.5836 $\pm$0.0103       |  0.0623$\pm$0.0008 |  0.2975$\pm$0.0028   | 0.4726$\pm$0.0092 |
> |Informer    |        0.1698 $\pm$0.0010       |    0.2647 $\pm$0.008     |  0.6165 $\pm$0.0395      | 0.0457$\pm$0.0005| 0.2599$\pm$0.0059  | 0.8749$\pm$0.0395|
> |DLinear    |        0.2529 $\pm$0.0066      |  0.2249 $\pm$0.0037    |  0.4537 $\pm$0.0031      |0.1401$\pm$0.0025|**0.1322**$\pm$0.0015 |0.5847$\pm$0.0038|
> |**VMFTransformer**    |       **0.0147** $\pm$0.0004     | **0.2018** $\pm$0.0090  |0.2583$\pm$0.0013   |**0.0136**$\pm$0.0024|0.2198$\pm$0.0085|**0.3029**$\pm$0.0009|
>
>
>
>
>
>
> |    Horizon = 24        |           | q-50-loss|       |       |  q-90-loss   |        |
> | :-----------------  | :----------------- | :----------------- | :-----------------  | :-----------------  | :-----------------  | :-----------------  |
> |**Model**  | Electricity    | Solar | Traffic | Electricity    | Solar | Traffic|
> |DeepAR    |       $\underline{0.0734} $$\pm$0.0027     | 0.4626 $\pm$0.0054 |$\underline{0.1526}$$\pm$0.0079    |0.0580$\pm$0.0021|0.3097$\pm$0.0118 |$\underline{0.1082}$$\pm$0.0014  |
> |SimpleFeedForward    |     0.0784 $\pm$0.0041    |0.5341$\pm$0.0047  |0.2518$\pm$0.0025    |$\underline{0.0433}$$\pm$0.0008|0.3591$\pm$0.0049|0.2011$\pm$0.0017  |
> |TFT    |     0.1107 $\pm$0.0055    |0.4855 $\pm$0.0113  |0.1981$\pm$0.00633    |0.0588$\pm$0.0014|0.2471$\pm$0.0122|0.1478$\pm$0.0020  |
> |Transformer    |     0.0789 $\pm$0.0126    |0.5354 $\pm$0.0107  |0.1644$\pm$0.0083    |0.0500$\pm$0.0012|0.3502$\pm$0.0167|0.1087$\pm$0.0029 |
> |Autoformer    |     0.1224 $\pm$0.0134   |$\underline{0.2233}$ $\pm$0.0946  |0.2156$\pm$0.0856    |0.0653$\pm$0.0023|$\underline{0.2112}$$\pm$0.0112|0.1454$\pm$0.0012|
> |PatchTST    |     0.1840 $\pm$0.0088    |0.5611 $\pm$0.0252  |0.4690$\pm$0.0297    |0.2291$\pm$0.0091|0.5678$\pm$0.1010|0.4367$\pm$0.0415|
> |Informer    |     0.2167 $\pm$0.0234    | 0.3456 $\pm$0.0617  |0.6055$\pm$0.0284   |0.1816$\pm$0.0294|0.2480$\pm$0.0181|0.6649$\pm$0.0151|
> |DLinear    |      0.1161 $\pm$0.0124   |0.2319$\pm$0.0062 |0.3418$\pm$0.0082   |0.1180$\pm$0.0141|0.2129$\pm$0.0051|0.3462$\pm$0.0085|
> |**VMFTransformer**   |     **0.0722** $\pm$0.0097  |**0.2125**$\pm$.0113|**0.1490**$\pm$0.0175|**0.0427**$\pm$0.0010 |**0.1371**$\pm$0.0346|**0.0935**$\pm$0.0089|

---

> > ### Author Response · Authors · 2023-11-23
> > **General response to all reviewers (part 3)**
> >
> > |    Horizon = 168         |           | q-50-loss|       |       |  q-90-loss   |        |
> > | :-----------------  | :----------------- | :----------------- | :-----------------  | :-----------------  | :-----------------  | :-----------------  |
> > |**Model**  | Electricity    | Solar | Traffic | Electricity    | Solar | Traffic|
> > |DeepAR    |         0.1463 $\pm$0.0061      |  0.5613 $\pm$0.0039  |0.1553$\pm$0.0081     |0.0862$\pm$0.0109|0.2349$\pm$0.0162  |**0.1012**$\pm$0.0018   |
> > |SimpleFeedForward    |      0.1397 $\pm$0.0022     |0.5890 $\pm$0.0107   |0.2348 $\pm$0.0034     |0.0730$\pm$0.0008|0.2557$\pm$0.0038 |0.1360$\pm$0.0020   |
> > |TFT    |      0.2242 $\pm$0.0497     | 0.6085 $\pm$0.0220   |$\underline{0.1514} $$\pm$0.0106     |0.1122$\pm$0.0050|0.2030$\pm$0.0662 |$\underline{0.1019}$$\pm$0.0122   |
> > |Transformer    |      0.1172 $\pm$0.0497     | 0.6318 $\pm$0.0220   |0.1739 $\pm$0.0106     |$\underline{0.0670}$$\pm$0.0050|0.3115$\pm$0.0662 |0.1063$\pm$0.0122  |
> > |Autoformer    |      0.1342 $\pm$0.0092    | $\underline{0.2354} $$\pm$0.0042   |0.2559 $\pm$0.0021     |0.0790$\pm$0.0024|$\underline{0.1588}$$\pm$0.0029 |0.1948$\pm$0.0086|
> > |PatchTST    |     $\underline{0.1171}$ $\pm$0.0007     | 0.3462 $\pm$0.0037   |0.3041 $\pm$0.005     |0.1189$\pm$0.0044|0.3976$\pm$0.0187 |0.3019$\pm$0.0094|
> > |Informer    |      0.3753 $\pm$0.0062     |  0.5434 $\pm$0.0083   |0.7194 $\pm$0.0028   |0.2440$\pm$0.0294|0.2960$\pm$0.0057|0.6800$\pm$0.0052|
> > |DLinear    |       0.1750 $\pm$0.0156    |0.2458 $\pm$0.0030  |0.3879 $\pm$0.0071    |0.1874$\pm$0.0033|0.2326$\pm$0.0051|0.3968$\pm$0.0083|
> > |**VMFTransformer**    |      **0.0839** $\pm$0.0042   |**0.2318**$\pm$0.0011 |**0.1225** $\pm$0.0099 |**0.0597**$\pm$0.0172|**0.1473**$\pm$0.0050|0.1099$\pm$0.0011|
> >
> >
> >
> >
> > |    Horizon = 720         |           | q-50-loss|       |       |  q-90-loss   |        |
> > | :-----------------  | :----------------- | :----------------- | :-----------------  | :-----------------  | :-----------------  | :-----------------  |
> > |**Model**  | Electricity    | Solar | Traffic | Electricity    | Solar | Traffic|
> > |DeepAR    |          0.1805 $\pm$0.0379       |   0.3543 $\pm$0.0095   | 0.2724$\pm$0.0330      | 0.1762$\pm$0.0659| $\underline{0.1451}$$\pm$0.0135   | 0.1883$\pm$0.0336    |
> > |SimpleFeedForward    |       0.1336 $\pm$0.0038      | 0.3738 $\pm$0.0271    | 0.2661 $\pm$0.0061      | 0.0854$\pm$0.0031| 0.1603$\pm$0.0239  | 0.1579$\pm$0.0018    |
> > |TFT    |       0.1748 $\pm$0.0055      |  0.3234 $\pm$0.0025    | $\underline{0.1906}$ $\pm$0.0061      | 0.0975$\pm$0.0091|**0.1084**$\pm$0.0061  | $\underline{0.1266}$$\pm$0.0052    |
> > |Transformer    |      $\underline{0.1246}$$\pm$0.0097      |  0.3463 $\pm$0.0172    | **0.1885** $\pm$0.0112      | $\underline{0.0829}$$\pm$0.0051| 0.1600$\pm$0.0091  |**0.1226**$\pm$0.0067   |
> > |Autoformer    |       0.3384 $\pm$0.0038      |  0.6827 $\pm$0.0067    | 0.3751 $\pm$0.0667      | 0.3138$\pm$0.0012| 0.5837$\pm$0.0038  |0.2969$\pm$0.0807|
> > |PatchTST    |       0.4222 $\pm$0.0013      |  0.5221 $\pm$0.0170    | 0.7957 $\pm$0.0171      | 0.2998$\pm$0.0034| 0.2723$\pm$0.0294  |0.4704$\pm$0.0218|
> > |Informer    |       0.3897 $\pm$0.0046      |   0.5601 $\pm$0.0245    | 0.8969 $\pm$0.0136     |0.2496$\pm$0.0049|0.3097$\pm$0.0029 |0.7540$\pm$0.0108|
> > |DLinear    |        0.2802 $\pm$0.0074     | **0.2598** $\pm$0.0043   | 0.6692 $\pm$0.0045     |0.3319$\pm$0.0138|0.2319$\pm$0.0014|0.7092$\pm$0.0054|
> > |**VMFTransformer**    |       **0.1028** $\pm$0.0028    |$\underline{0.2556} $$\pm$0.0242  |0.5738$\pm$0.0074  |**0.0798**$\pm$0.0288|0.5455$\pm$0.0438|0.3917$\pm$0.0032|

---

> > > ### Author Response · Authors · 2023-11-23
> > > **General response to all reviewers (part 4)**
> > >
> > > A sample of 50,000 data points was selected from the Solar dataset to test the time consumed by our model across different horizons.
> > > |Encoding length	|Time consumed (seconds)|
> > > | :-----------------  | :----------------- |
> > >  | 12	  | 85.352 |
> > >  | 48	  | 122.309 |
> > >  | 72	  | 249.441 |
> > >  | 120 | 	 444.464 |
> > >  | 168 | 	 1629.578 |
> > >  | 240 | 	 2699.469 |
> > >  | 360 | 	6767.158 |
> > >
> > > A sample of 50,000 data points was selected from the Solar dataset to test the memory consumed by our model across different horizons.
> > > |Encoding length	|Memory usage (Byte)|
> > > | :-----------------  | :----------------- |
> > >  |12	  |2498048 |
> > >  |48	 | 3061248 |
> > >  |72	 | 3462656 |
> > >  |120	 | 4314624 |
> > >   |168	 | 5248512 |
> > >  |240	 | 6776320 |
> > >  |360	 |9700864 |

---

> > > > ### Author Response · Authors · 2023-11-23
> > > > **General response to all reviewers (part 5)**
> > > >
> > > > We conducted additional ablation studies to prove the effectiveness of our designed Angle & Scale similarity measurements in the attention module when used in conjunction with our proposed VMF-Loss. However, we found that these similarity measurements were not as effective when the model was trained using only the standard MSE-loss. This suggests that our Angle & Scale similarity design has a beneficial interaction specifically with the VMF-Loss function during training. To further analyze this relationship, we performed experiments altering components of both the loss function and attention module. We found that removing either the angle or scale measurement from the attention with VMF-Loss hurt performance, confirming that both aspects contribute to its effectiveness under VMF-Loss. Furthermore, incorporating only one of the two measurements along with MSE-Loss did not lead to significant improvements, verifying that the combination of both measurements works synergistically with properties unique to the VMF-Loss function.
> > > >
> > > > |VMF-Loss|	Metric	|Angle&Scale	|Dot-product	|
> > > > | :-----------------  | :----------------- |  :-----------------  | :----------------- |
> > > > |Horizon=24	|MAE|	**0.2339** |	0.2389 |
> > > > |	|MSE|	0.1828 |	**0.1749** |
> > > > |	|50-loss|	**0.3238** |	0.3317 |
> > > > |	|90-loss|	**0.2667** |	0.2707 |
> > > > |Horizon=168	|MAE|	**0.2014** |	0.2389 |
> > > > |	|MSE|	**0.1307** |	0.1316 |
> > > > |	|50-loss|	**0.2848** |	0.2950 |
> > > > |	|90-loss|	**0.1724** |	0.2250 |
> > > >
> > > > |MSE-Loss	|Metric|	Angle&Scale|	Dot-product|
> > > > | :-----------------  | :----------------- |  :-----------------  | :----------------- |
> > > > |Horizon=24	|MAE	|0.7389 	|**0.7179** |
> > > > |	|MSE	|**0.4792** 	|0.5061 |
> > > > |	|50-loss	|0.5059 	|**0.5010** |
> > > > |	|90-loss	|**0.8153** 	|0.8650 |
> > > > |Horizon=168|MAE	|0.7179 	|0.7179 |
> > > > |	|MSE|	0.5075 |	**0.5061** |
> > > > |	|50-loss|	0.5010 |	0.5010 |
> > > > |	|90-loss|	**0.8631** |	0.8685 |

---

> > > > > ### Author Response · Authors · 2023-11-23
> > > > > **General response to all reviewers (part 6)**
> > > > >
> > > > > **Learning curves (Figure 3)** of our model are provided in the updated version to demonstrate the stability of the training procedure.
> > > > >
> > > > > **Comparison of metrics corresponding to different horizons where H ranges from 1 to 720 (Figure 4)** of our model is provided in the updated version to demonstrate the stability of forecasting performance.

---

### Meta-Review · Area_Chair_5mJC · 2023-12-07

**Metareview:**

The reviewers raised multiple concerns, notably the experimental setups. After the rebuttal, the reviewers acknowledge that there is still room for improvement, and I agree with their expertise. Nevertheless, the angle and scaling idea is interesting, and the authors are encouraged to continue working towards this direction.

**Justification For Why Not Higher Score:**

The experimental setups are not good.

**Justification For Why Not Lower Score:**

n/a

---

### Decision · Program_Chairs · 2024-01-16

Reject